# The Positive Role of Nanometric Molybdenum–Vanadium Carbides in Mitigating Hydrogen Embrittlement in Structural Steels

**DOI:** 10.3390/ma14237269

**Published:** 2021-11-28

**Authors:** Luis Borja Peral, Inés Fernández-Pariente, Chiara Colombo, Cristina Rodríguez, Javier Belzunce

**Affiliations:** 1Department of Material Science and Metallurgical Engineering, Polytechnic School of Engineering of Gijón, University of Oviedo, 33203 Gijón, Spain; inesfp@uniovi.es (I.F.-P.); cristina@uniovi.es (C.R.); belzunce@uniovi.es (J.B.); 2Department of Mechanical Engineering, Polytechnic of Milan, 20133 Milano, Italy; chiara.colombo@polimi.it

**Keywords:** TDA analysis, electrochemical hydrogen permeation, fracture toughness, fatigue crack growth rate, nanometric (Mo,V)C

## Abstract

The influence of hydrogen on the fracture toughness and fatigue crack propagation rate of two structural steel grades, with and without vanadium, was evaluated by means of tests performed on thermally precharged samples in a hydrogen reactor at 195 bar and 450 °C for 21 h. The degradation of the mechanical properties was directly correlated with the interaction between hydrogen atoms and the steel microstructure. A LECO DH603 hydrogen analyzer was used to study the activation energies of the different microstructural trapping sites, and also to study the hydrogen eggresion kinetics at room temperature. The electrochemical hydrogen permeation technique was employed to estimate the apparent hydrogen diffusion coefficient. Under the mentioned hydrogen precharging conditions, a very high hydrogen concentration was introduced within the V-added steel (4.3 ppm). The V-added grade had stronger trapping sites and much lower apparent diffusion coefficient. Hydrogen embrittlement susceptibility increased significantly due to the presence of internal hydrogen in the V-free steel in comparison with tests carried out in the uncharged condition. However, the V-added steel grade (+0.31%V) was less sensitive to hydrogen embrittlement. This fact was ascribed to the positive effect of the precipitated nanometric (Mo,V)C to alleviate hydrogen embrittlement. Mixed nanometric (Mo,V)C might be considered to be nondiffusible hydrogen-trapping sites, in view of their strong hydrogen-trapping capability (~35 kJ/mol). Hence, mechanical behavior of the V-added grade in the presence of internal hydrogen was notably improved.

## 1. Introduction

For contributing to the development of the impending new green economy society based on the use of CO_2_-free alternative energy sources, the world’s energy industries must be decarbonized [1]. In this respect, hydrogen energy is postulated as an attractive alternative for an immediate future, in which the world’s population is expected to grow to 10 billion people by the year 2050 [2]. Hence, it is important to contribute to the development of structural steels for the manufacture of vessels and pipelines able to store and transport hydrogen at high working pressures, up to 70 or even 100 MPa, in order to accommodate the imminent increase in energy demand.

It is well known that hydrogen atoms can penetrate and diffuse within the steel microstructure, deteriorating the steel’s mechanical properties under hydrogen environments. In this respect, different authors have reported a notable decrease in tensile strength and ductility due to the effects of hydrogen [3,4,5,6]. Hydrogen embrittlement susceptibility increases with the strength level, and also is highly dependent on the applied displacement rate.

The influence of hydrogen on the fracture toughness has not gone unnoticed by the scientific community [7,8,9,10]. In this same context, Ogawa et al. [11] analyzed the effect of hydrogen gas under the pressures of 0.7 and 115 MPa on a low-carbon steel (σ_y_ = 360 MPa). They reported a strong decrease in J_IC_, from ~350 kJ/m^2^ in air to ~75 kJ/m^2^ under 115 MPa of pressure. However, hydrogen’s effect was slight at 0.7 MPa. On the other hand, Colombo et al. [12] studied the effects of internal hydrogen in electrochemically precharged AISI4130 steel samples (σ_y_ = 715 MPa). They also reported a strong decrease in J_IC_, from 215.5 kJ/m^2^ in air to 22 kJ/m^2^ in the presence of internal hydrogen.

Several authors have also reported the influence of hydrogen on the fatigue crack growth rate [13,14,15]. The effect of hydrogen on the fatigue crack growth performance of a pure iron steel was investigated by T. Shinko et al. [16]. Tests were directly conducted on hydrogen gas, under a pressure of 35 MPa and at different test frequencies. Two different propagation regimes were identified. The first one, at low ΔK values, was characterized by a moderated fatigue crack growth enhancement, whilst for ΔK > 14 MPa·m^0.5^, hydrogen gas caused an acceleration factor in the crack growth rate approximately 10–30 times higher than in air. A. Álvaro et al. [17] studied the impact of hydrogen on the fatigue crack growth rate of a X70 pipeline steel grade (σ_y_ = 485 MPa). In this study, fatigue tests were carried out in situ using cathodically charged samples. The fatigue crack growth rate was notably influenced by the test frequency. At the lowest test frequency (0.1 Hz), hydrogen induced an increase in the crack growth rate of about 100 times compared to testing in air.

Therefore, vessels and pipelines used to store and transport hydrogen must be able to provide a safe service for long periods of time, being essential to ensure good resistance to the hydrogen embrittlement phenomenon. In this respect, quenched and tempered steels alloyed with chromium, chromium–molybdenum, or chromium–molybdenum–vanadium are often employed in these type of facilities [18]. However, hydrogen atoms, motivated by the existing high hydrostatic stress in microcracks or notches, can diffuse toward a damage process zone (Figure 1a), promoting hydrogen embrittlement when a critical hydrogen concentration is reached [19].

On the other hand, microstructural heterogeneities, also known as ‘hydrogen trapping sites’, also play an important role in preventing hydrogen embrittlement. Therefore, it is important to determine preferential hydrogen trapping sites [20,21,22]. Accordingly, high energy traps, uniformly distributed within the steel microstructure, might notably contribute to delaying hydrogen diffusion toward the aforementioned damage process zone (Figure 1b) in order to improve mechanical performance in hydrogen environments. For instance, the addition of carbide-forming elements (V, Ti, or Nb) was demonstrated to be effective in relieving hydrogen embrittlement susceptibility [23,24,25].

A reversible or irreversible trapping character [20,26] is associated with the activation energy of hydrogen atoms detrapping from the different microstructural singularities, and it will have an important impact on the mechanical behavior of the steel in fighting against hydrogen embrittlement [27,28]

In this study, the influence of the vanadium addition (+0.31%) on the fracture toughness and fatigue crack growth rate performance of samples thermally precharged with hydrogen was evaluated. The influence of hydrogen on the mechanical properties’ deterioration was directly correlated with the hydrogen–microstructure interaction.

## 2. Experimental Procedure

### 2.1. Materials, Heat Treatments, and Samples

Two different low-alloyed ferritic steels were selected in this study: with and without vanadium. The chemical composition of the steels, in weight %, is shown in Table 1.

Regarding the applied heat treatments, the V-free steel was austenitized at 940 °C for 3 h, then it was quenched in water (WQ) and tempered at 690 °C for 30 h. The V-added steel was austenitized at 925 °C for 90 min, quenched in water, and tempered at 720 °C for 3 h. The sequence of heat treatments and the nomenclature of the obtained steel grades are shown in Table 2.

To study the interaction between hydrogen atoms and the steel microstructure, small cylindrical samples (10 mm diameter) were employed to measure the introduced hydrogen content and the kinetics of hydrogen departure at room temperature (RT). The same geometry was employed to determine the binding energies of microstructural hydrogen traps. Furthermore, the electrochemical hydrogen permeation technique was applied to determine the apparent hydrogen diffusion coefficient. Samples with a 1 mm thickness were used.

On the other hand, to study the influence of internal hydrogen on the mechanical properties, fracture toughness and fatigue crack growth rate characterization were determined using compact test (CT) samples.

### 2.2. Hydrogen Precharging Methodology

All the samples were precharged with gaseous hydrogen in a high-pressure hydrogen reactor that was manufactured in accordance with the ASTM G146 standard [29]. The hydrogen precharging conditions are listed in Table 3.

After the mentioned 21 h at 450 °C, a cooling phase of 1 h, until reaching a temperature of 85 °C (keeping the hydrogen pressure at 19.5 MPa to limit hydrogen departure) was always used for removing the samples from the hydrogen reactor. Consequently, to minimize the hydrogen losses, all the samples were conserved in liquid nitrogen at −196 °C until testing.

### 2.3. Hydrogen Desorption at Room Temperature

The hydrogen egression kinetics at room temperature was determined. All the cylindrical hydrogen-precharged pins (10 mm diameter and 20 mm length) were removed from the liquid nitrogen simultaneously. Then, they were exposed to air at room temperature in different time intervals, and finally samples were introduced in a LECO DH603 hydrogen analyzer in order to measure the hydrogen concentration. The thermal analysis to determine hydrogen concentration consisted of maintaining the samples at 1100 °C for 400 s [30].

### 2.4. Trap Binding Energies

In order to determine the activation energy (E_a_) of hydrogen traps present in the microstructure of the studied steel grades, hydrogen desorption profiles were carried out in the hydrogen LECO DH603 analyzer under different heating rates: 3600, 2400, 1800, and 1200 °C/h [30].

### 2.5. Electrochemical Hydrogen Permeation Technique

Hydrogen absorption and diffusion were analyzed using an electrochemical double cell, as previously described in [26]. A circular area of approximately 1 cm^2^ was exposed to the solution, and the electrochemical tests were conducted at room temperature (±20 °C). Hydrogen atoms were generated on the charging cell under a current density of 1 mA/cm^2^, using a 2 mol/L H_2_SO_4_ + 0.25 g/L of As_2_O_3_ solution (pH = 1). The hydrogen exit side contained 0.1 mol/L NaOH solution (pH = 12).

The apparent hydrogen diffusion coefficient (D_app_) and subsurface hydrogen concentration at the entry side (C_app_) were calculated following the standard [31].

### 2.6. Tensile Properties and Hardness

To determine the fundamental mechanical properties of the selected grades of steel, tensile tests were performed in air (without hydrogen) on an Instron 5582 tensile testing machine. Smooth specimens with a diameter of 5 mm and a calibrated length of 28 mm were used [32]. A displacement rate of 0.4 mm/min was employed.

Brinell hardness (HB) was also calculated using a Hoytom hardness tester, using a load of 187.5 kg and a ball 2.5 mm in diameter [33].

### 2.7. Fracture Toughness Tests

Fracture toughness characterization was performed using compact test (CT) specimens with a 48 mm width and a 12 mm thickness. Fracture toughness tests were carried out following the ASTM E1820 standard [34].

First of all, fracture toughness tests, without hydrogen, were carried out under a nominal displacement rate of 1 mm/min. Then, hydrogen-precharged samples were tested under two different displacement rates, 1 and 0.01 mm/min, in order to study the impact of hydrogen on the fracture toughness. The fracture toughness initiation parameter ‘J_0.2_’ (kJ/m^2^) was used in this study. This parameter corresponded to the value of ‘J’ after a crack growth of 0.2 mm offset from the blunting line.

### 2.8. Fatigue Crack Growth Tests

The fatigue crack growth rate also was determined using compact test (CT) samples, with a width of 48 mm and a thickness of 10 mm, following the ASTM E647 standard [35].

In order to determine the fatigue crack growth behaviour of each steel grade without internal hydrogen, uncharged samples were tested at room temperature under a load ratio R = 0.1 and a frequency of 10 Hz. Consequently, fatigue crack growth tests were carried out on hydrogen-precharged samples [36]. All the tests were conducted in air, at room temperature, under a load ratio R = 0.1. Test frequencies of 10, 1, and 0.1 Hz were employed in order to evaluate the influence of this parameter on the fatigue crack growth rate performance. Hence, the curves representing the fatigue crack growth rate (da/dN) versus the stress intensity factor range (ΔK) were obtained under the different test conditions.

### 2.9. SEM and TEM Analysis

The microstructures were observed in a scanning electron microscope (SEM JEOL-JSM5600) using an acceleration voltage of 20 kV. Initially, the samples were ground (up to 1200 SiC paper), and then samples were polished with 6 and 1 µm diamond paste, respectively. Finally, they were etched with Nital-2%.

On the other hand, TEM analysis was also performed on the V-free and V-added steel grades. A JEOL JEM-2100F field-emission transmission electron microscope, operating at an acceleration voltage of 200 kV, was employed. Sample preparation was as described in [26].

## 3. Results

### 3.1. Microstructures

The obtained microstructures of the V-free and V-added steels after the heat treatments described in Table 2 are shown in Figure 2. Figure 2A,B reveal the tempered martensite present on both grades of steel. The prior austenite grain size was around 25 µm.

The profuse precipitation of carbides, which took place during the tempering treatment, can be clearly seen along the prior austenite grain boundaries, and also within the grain. In this respect, Fe–Cr–Mo mixed carbides, such as M_7_C_3_, M_2_C, and M_23_C_6_, were identified in the V-free steel grade by means of the TEM analysis, whilst mixed carbides of V and Mo with a finer particle size were identified in the V-added steel (Figure 2—C1, C2 and C3).

### 3.2. Mechanical Properties

Tensile properties and Brinell hardness level, obtained as described in Section 2.6, are summarized in Table 4.

### 3.3. Hydrogen Desorption at Room Temperature

Hydrogen concentration measured as a function of the exposure time of the cylindrical samples at room temperature (RT) is shown in Table 5. Although more hydrogen (4.3 ppm) was introduced into the V-added steel grade, the percentage of hydrogen that was able to emerge from traps and diffuse out of the steel samples after 48–72 h was notably lower (~5%). According to this, we postulated that the V-added steel grade presented more hydrogen-trapping capability (i.e., high-energy trapping sites for hydrogen atoms).

### 3.4. Trap Binding Energies

As an example, Figure 3a shows the hydrogen thermal desorption profiles obtained in the V-free steel grade under the different heating rates, from 1200 to 3600 °C/h. In order to appreciate the effect of V-addition (+0.31%) on hydrogen trapping, Figure 3b displays the hydrogen desorption profiles under a heating rate of 1800 °C/h for both steel grades. In this respect, the hydrogen desorption profile was shifted to a higher temperature due to the V-addition. As will be shown below, this phenomenon was associated with the precipitated nanometric molybdenum–vanadium carbides, which seemed to act as strong traps for hydrogen diffusion.

Lastly, Figure 3c gives the linear regressions applied to determine the detrapping activation energies (E_a_) related to the different peaks, which were identified both in the V-free steel and in the V-added grade.

Based on the obtained results, two desorption peaks were identified for both steel grades.

Regarding the V-free steel grade, the first peak showed an activation energy of 13 kJ/mol that was associated with hydrogen detrapping from (Cr,Mo) carbides [37]. Nevertheless, the second desorption peak, the activation energy of which was 18 kJ/mol, was attributed to the presence of lath and packet martensite interfaces [38,39]. Both trapping sites were considered as reversible traps for hydrogen atoms. The trap activation energies are summarized in Table 6.

On the other hand, due to the V-addition (+0.31%), the second peak was notably shifted to a higher temperature (Figure 2B). According to this, the highest activation energy, ~35 kJ/mol, was obtained in the V-added grade. This energy was associated with hydrogen detrapping from the aforementioned precipitated nanometric molybdenum–vanadium carbides (10–30 nm, Figure 2C1), and agreed well with the previous works in [40,41,42].

This fact also contributed to reinforcing the results previously shown in Table 5. Here, in the V-added grade, around 95% of the hydrogen remained strongly trapped in the steel microstructure. This fact confirmed that nanometric molybdenum–vanadium carbides (35 kJ/mol) could be considered, in this study, as irreversible trapping sites. These carbides were previously identified in Figure 2C by means of the TEM analysis.

### 3.5. Electrochemical Hydrogen Permeation Technique

Figure 4 shows the hydrogen permeation curves obtained for both grades of steel: with and without vanadium. The apparent permeability (P_app_), the apparent hydrogen concentration (C_app_), and the apparent diffusion coefficient (D_app_) that were extracted from the hydrogen permeation curves are presented in Table 7.

It is important to note that hydrogen diffusion kinetics were notably delayed due to the vanadium addition (+0.31%). This fact was associated with the stronger trapping sites previously identified in the V-added steel grade, (V,Mo)C.

### 3.6. Fracture Toughness Tests

The load-COD displacement curves and the fracture toughness crack growth (J-R) curves, corresponding to the V-free and V-added steel grades, are respectively shown in Figure 5 and Figure 6. In the same figures, the effect of the internal hydrogen can be clearly appreciated at the different displacement rates (1–0.01 mm/min) by comparing them with the uncharged sample (1 mm/min).

Using the results given in Figure 5 and Figure 6, the fracture toughness initiation parameter (J_0.2_) and the embrittlement indexes (EI) were determined, and are reported in Table 8. In the same table, the fracture micromechanisms, observed in the SEM analysis, have been also indicated by order of importance.

Based on the obtained results, the internal hydrogen effect was notable in these grades of steel. It is important to note that the effect of the internal hydrogen increased at the lowest displacement rates (0.1 mm/min), and this effect was more notable in the V-free steel grade, in which embrittlement indexes close to 50% were calculated. Accordingly, the fracture mechanism was modified in the presence of internal hydrogen (Table 8). However, in the V-added steel grade, microvoid coalescence (MVC) was the only micromechanism identified on the fracture surfaces, even in the presence of internal hydrogen, and also for the lowest displacement rates.

### 3.7. Fatigue Crack Growth Rate Tests

Figure 7 shows the fatigue crack growth curves obtained for the V-free and V-added steel grades in the uncharged condition (R = 0.1 and 10 Hz). In the same figure, these curves are compared with those determined for the hydrogen-precharged samples and tested at the different frequencies (1 and 0.1 Hz).

The acceleration effect on the fatigue crack growth rate, due to the internal hydrogen effect, was only observed in the V-free steel grade. In this grade of steel, the hydrogen effect was observed for the lowest test frequency (0.1 Hz), and it was especially marked at the lowest ΔK values. According to this, at around ΔK = 30 MPa·m^0.5^ and for a test frequency of 0.1 Hz, the fatigue crack growth rate was around 15 times quicker than that determined in the uncharged sample. At 0.1 Hz, hydrogen diffusion was coupled with the crack growth rate (based on its diffusion coefficient, 1.7 × 10^−10^ m^2^/s), and consequently, hydrogen atoms diffused in and out of the process zone in every fatigue cycle, thereby enabling the existence of the so-called embrittled process zone due to hydrogen accumulation (Figure 1). This fact would explain the increase in the crack growth rate observed in the V-free grade. Similar results were reported in [17,44,45,46]. Hydrogen’s impact on the fatigue crack growth acceleration factor can be clearly appreciated in Figure 8 under the different applied test frequencies, and also for the different ΔK values (30–50 MPa·m^0.5^).

On the other hand, the internal hydrogen effect was totally negligible in the V-added steel grade, even for the lowest applied test frequency (0.1 Hz, Figure 7). The microstructure of this steel grade presented uniform dispersion of nanometric molybdenum–vanadium carbides, which behaved as very strong hydrogen trapping sites, thereby greatly reducing its diffusivity (D_app_ = 2.6 × 10^−11^ m^2^/s and H_D_ = 5%). Hence, local hydrogen concentration in the notch tip region was not able to give rise to hydrogen embrittlement.

## 4. Discussion

In order to analyze the effect of the steel microstructure on the hydrogen embrittlement phenomenon, Figure 9 shows a comparison of the apparent diffusion coefficient (D_app_) and the activation energies (E_a_) from the identified trapping sites in the studied steel grades, with and without vanadium.

Due to the addition of vanadium (+0.31%), the apparent diffusion coefficient (D_app_) in the V-added grade (2.6 × 10^−11^ m^2^/s) was around seven times lower than that estimated in the V-free grade (1.7 × 10^−10^ m^2^/s). This behavior was directly associated with the activation energy of the stronger trapping sites for hydrogen atoms identified in the V-added steel grade ~35 kJ/mol (desorption from the molybdenum–vanadium mixed nanometric carbides). However, weaker trapping sites were found in the V-free grade (Figure 9). Activation energy for hydrogen detrapping from nanometric (Mo,V)C (in the V-added grade) was well above the activation energy for detrapping from (Cr,Mo)C previously identified in the V-free grade (~13 kJ/mol).

Focusing now on the mechanical behavior in presence of internal hydrogen, the embrittlement indexes (EIs) estimated in the V-added steel from the fracture toughness tests were clearly lower than that calculated in the V-free grade, independently of the applied displacement rate (Figure 10). Hydrogen–microstructure interaction (Figure 9) explains this fact. As mentioned, the V-added grade showed a strong hydrogen-trapping capability, which contributed to retarding hydrogen diffusion toward the notch tip region (i.e., the process area). This fact is responsible for the mechanical behavior improvement observed in the V-added grade in presence of internal hydrogen. Despite having absorbed a high hydrogen concentration, around 4.3 ppm (Table 5), only 5% of such hydrogen was able to diffuse (~0.2 ppm) within the steel microstructure to reach the process zone, and this local concentration was barely able to trigger hydrogen embrittlement mechanisms (HEDE/HELP). Accordingly, a fully ductile fracture micromechanism (Figure 11b) was noticed in all the fracture toughness tests (even with hydrogen).

Nevertheless, in the case of the V-free steel, with weaker trapping sites, diffusible hydrogen (~67%, ~0.4 ppm) moved and accumulated in the process zone ahead of the crack tip of the CT sample, giving rise to a change in the failure micromechanism, from ductile (without hydrogen) to decohesion of packet and block martensite boundaries, the plasticity-related hydrogen-induced cracking (PRHIC) micromechanism, in the presence of internal hydrogen (Figure 11a).

The effect of the internal hydrogen on the fatigue performance of the V-free steel was especially notable (Figure 12). For the lowest test frequency (0.1 Hz), hydrogen atoms had more time to diffuse and attain the process zone, leading to embrittlement mechanisms (PRHIC and intergranular fracture in Figure 13a). However, as ∆K increased, the hydrogen effect seemed to diminish. In this case, the crack propagated faster than the hydrogen diffusion rate toward the crack tip (i.e., the hydrogen accumulation decreased), and consequently, the acceleration rate was alleviated. A similar effect was also reported in [12].

On the other hand, due to the V-addition, the V-added steel grade performance in the presence of internal hydrogen was greatly improved, and the effect of the test frequency was totally negligible, even for the lowest test frequency (Figure 12). Ductile fatigue striations (white circles in Figure 13b) were frequently observed in the V-added grade, both in the precharged sample and the uncharged one.

Mixed molybdenum–vanadium nanometric carbides definitely acted as very strong trapping sites for hydrogen atoms with an irreversible trapping character, and consequently, the local hydrogen concentration in the notch tip region was barely able to trigger hydrogen embrittlement due to V-addition.

It is important to recall that in the V-added grade, the fracture mechanism was always ductile. It was characterized by the presence of MVC in the fracture toughness tests, even with hydrogen, and by the presence of ductile fatigue striations in the fatigue tests, also in the presence of internal hydrogen.

Finally, it is important to highlight that, although several authors have reported that hydrogen embrittlement susceptibility increases as yield strength also increases [3,4,5,7]; in our study, the V-added steel grade (σ_y_ = 567 MPa), with a higher yield strength than the V-free grade (σ_y_ = 430 MPa), was less sensitive to the effects of hydrogen. Hence, we can emphasize that depending on the trap morphology for hydrogen atoms, the steel strength level and the total hydrogen content can be independent of the embrittlement level.

## 5. Conclusions

Based on the results from a wide experimental study, several conclusions were drawn:The V-added steel grade absorbed much hydrogen compared to the V-free steel, due to the fact that mixed molybdenum–vanadium nanometric carbides are very strong hydrogen trapping sites.Two hydrogen traps, with activation energies of 13 kJ/mol and 18 kJ/mol, were detected in the V-free steel grade. Stronger trapping sites were identified in the V-added grade (+0.31%), with an activation energy around 35 kJ/mol. This high activation energy was attributed to the presence of mixed molybdenum–vanadium nanometric carbides precipitated during the tempering treatment. Diffusible hydrogen and the hydrogen apparent diffusion coefficient were much lower in the V-added grade.In the V-free steel grade, J_0.2_ notably decreased due to the effects of the internal hydrogen. Embrittlement indexes in the order of 50% were found. This embrittlement was consistent with the change appreciated in the fracture micromechanisms: A PRHIC mechanism was identified in some areas of the precharged samples, while MVC was the present mechanism in the uncharged ones. However, lower embrittlement indexes (19–33%) were obtained in the V-added grade. Even with internal hydrogen, the V-added steel grade did not modify the failure micromechanism with respect to that observed on the uncharged samples (MVC).The fatigue crack propagation rate strongly increased in the V-free grade. Hydrogen damage was especially notable at the lowest test frequency (0.1 Hz) and lowest ΔK values. Here, the failure micromechanism changed from ductile fatigue striations (uncharged samples) to PRHIC and intergranular fracture (precharged samples). Nevertheless, the V-added grade did not show signs of hydrogen embrittlement, even at the lowest test frequency. A ductile fatigue striation micromechanism was observed in all the tested samples.The fracture toughness and fatigue behavior of the V-added steel grade in the presence of internal hydrogen was greatly improved, as the microstructure of this steel grade presented uniform dispersion of mixed molybdenum–vanadium nanometric carbides, which behaved as very strong trapping sites for hydrogen atoms. Despite having absorbed 4.3 ppm (versus 0.6 ppm absorbed in the V-free grade), most of the hydrogen remained strongly trapped into the microstructure (~95%). This fact contributed to limiting hydrogen accumulation in the crack-tip region, contributing to improved mechanical performance in hydrogen environments.After this wide study, it is possible to affirm that hydrogen embrittlement does not depend on the absorbed hydrogen concentration.The V-addition (+0.31%) demonstrates that the increase of the yield strength from 430 to 567 MPa did not increase the hydrogen embrittlement susceptibility. The type of trap must be taken into consideration to analyze the deterioration of mechanical properties in the presence of hydrogen.

## Figures and Tables

**Figure 1 materials-14-07269-f001:**
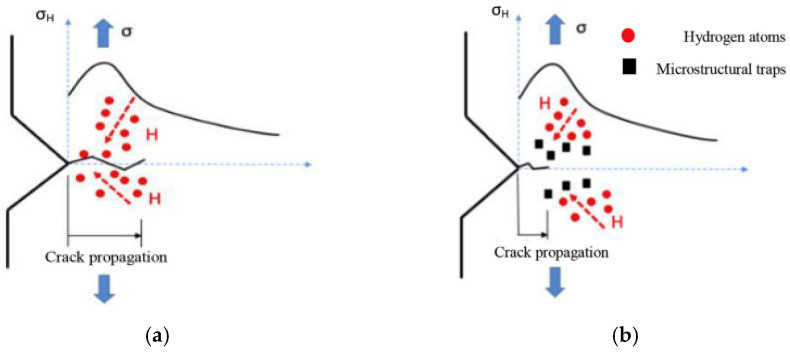
Hydrogen accumulation in the embrittlement process area (notched region) under a high hydrostatic stress. (**a**) Without traps. (**b**) With traps.

**Figure 2 materials-14-07269-f002:**
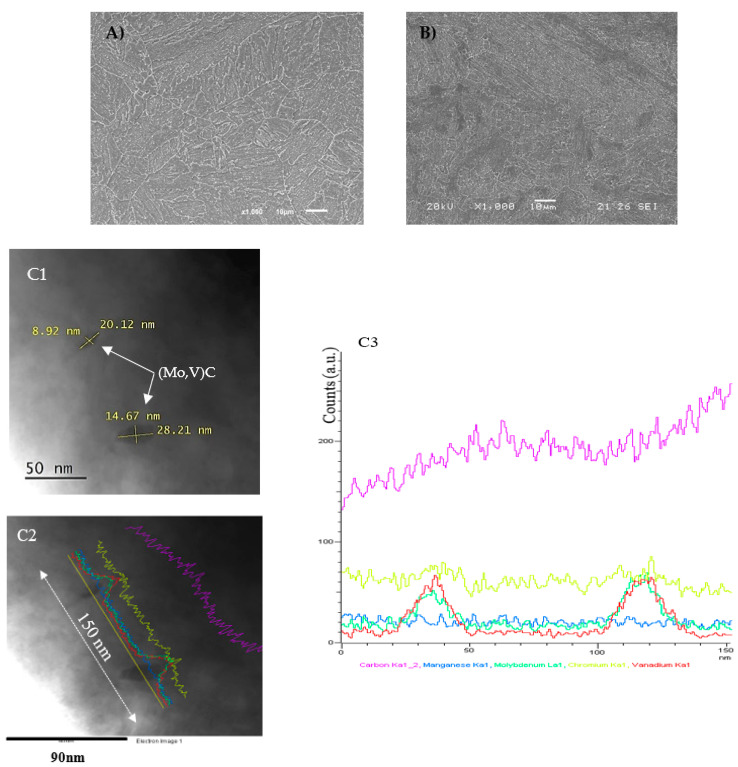
(**A**) Tempered microstructure corresponding to V-free steel; (**B**) tempered microstructure corresponding to V-added steel; and (**C1**–**C3**) TEM analysis of the V-added grade. (**C1**,**C2**) STEM images; the yellow line in (**C2**) shows the element line-scan profiles for C (purple), Cr (lime), V (red), Mo (green), and Mn (blue). (**C3**) Number of counts versus distance recorded along the yellow line (150 nm) shown in the image in (**C2**).

**Figure 3 materials-14-07269-f003:**
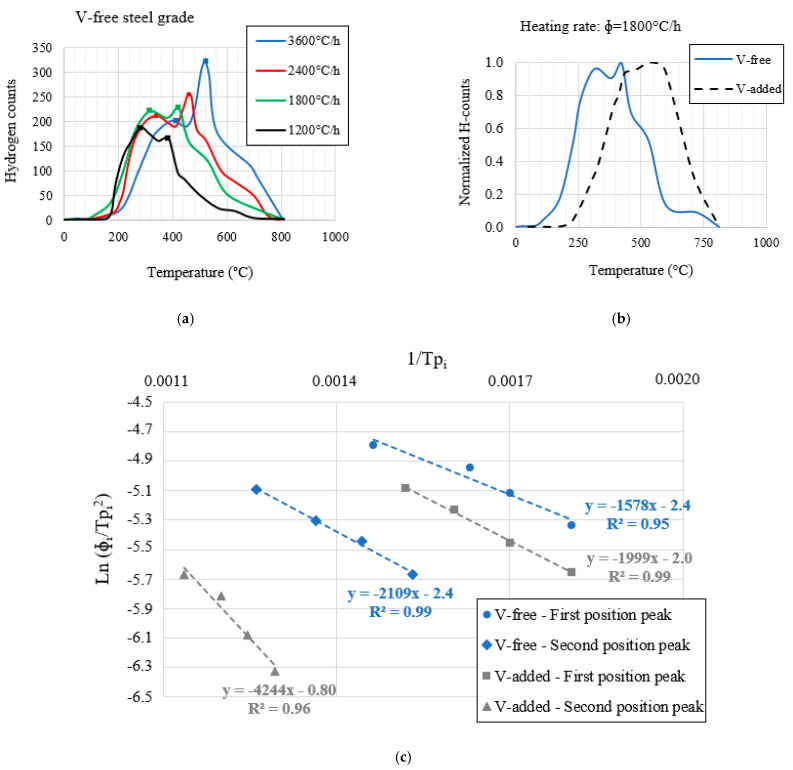
(**a**) Hydrogen thermal desorption profiles for the V-free steel grade; (**b**) comparison of hydrogen thermal desorption profiles at 1800 °C/h; and (**c**) estimation of trap binding energies.

**Figure 4 materials-14-07269-f004:**
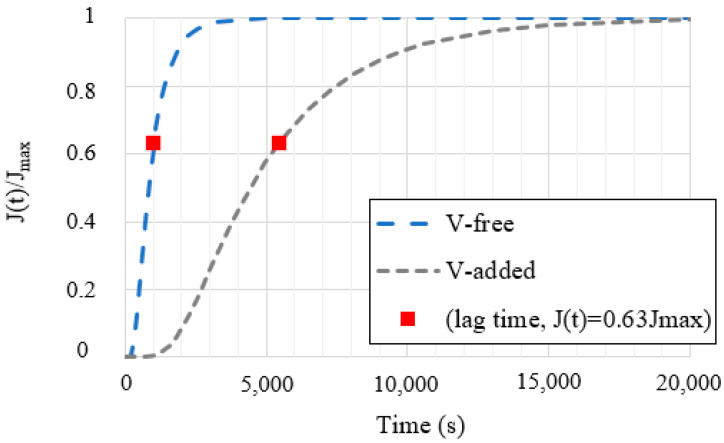
Electrochemical hydrogen permeation curves.

**Figure 5 materials-14-07269-f005:**
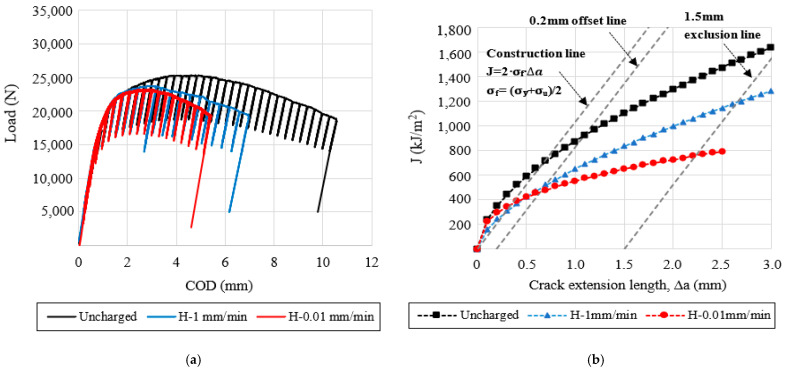
Hydrogen effect on the fracture toughness of the V-free steel grade tested at different displacement rates: (**a**) load-COD curves; (**b**) J-R curves.

**Figure 6 materials-14-07269-f006:**
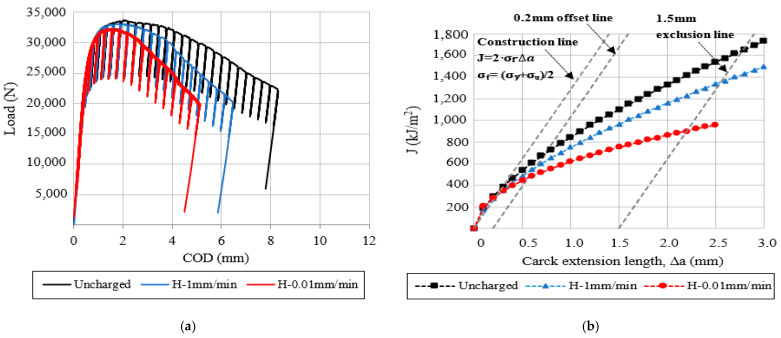
Hydrogen effect on the fracture toughness of the V-added steel grade tested at different displacement rates: (**a**) load-COD curves; (**b**) J-R curves.

**Figure 7 materials-14-07269-f007:**
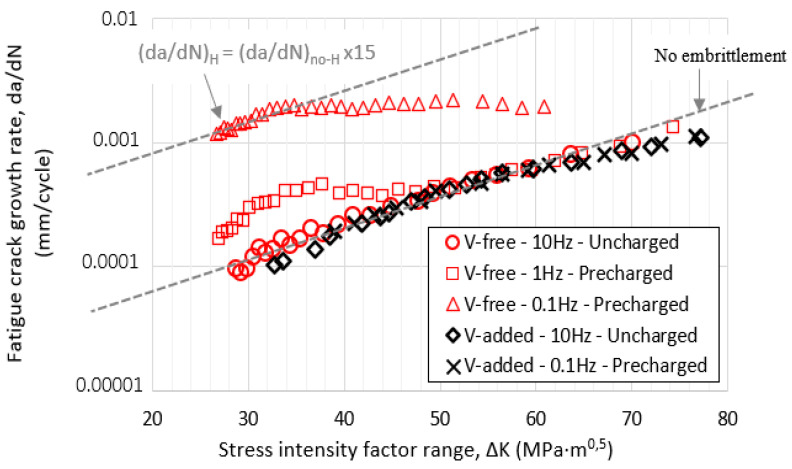
The da/dN versus ΔK curves of the studied steel grades at R = 0.1 and at different test frequencies (from 10 to 0.1 Hz).

**Figure 8 materials-14-07269-f008:**
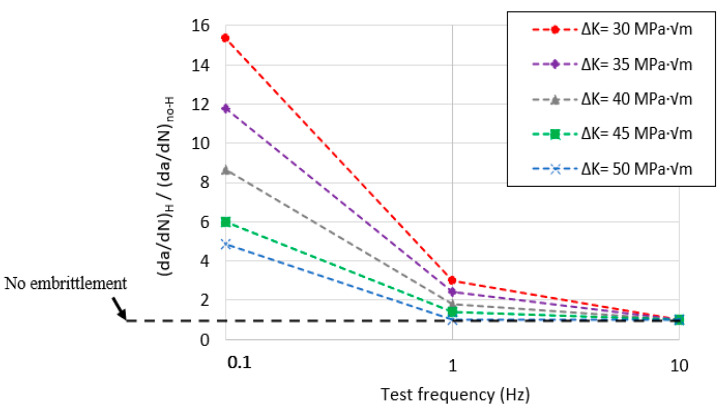
Fatigue crack growth acceleration factor in the V-free steel grade (with R = 0.1).

**Figure 9 materials-14-07269-f009:**
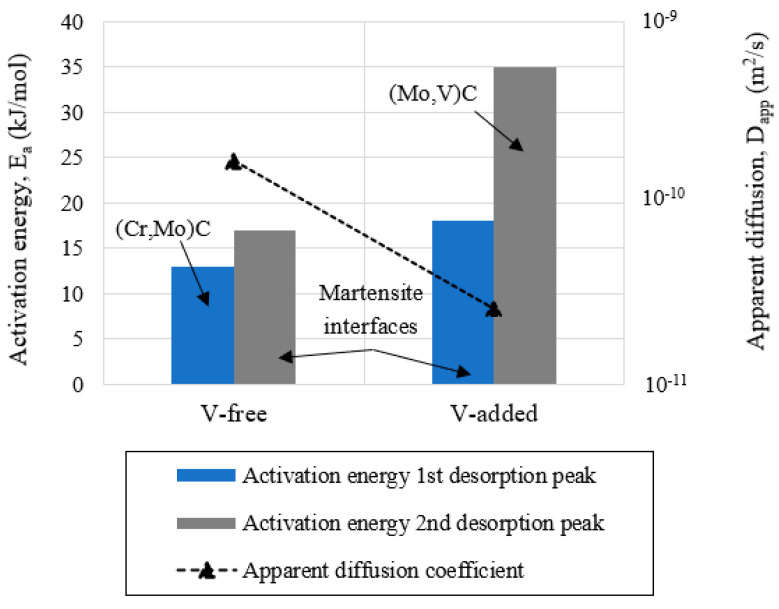
Hydrogen trapping and diffusion for V-free and V-added grades.

**Figure 10 materials-14-07269-f010:**
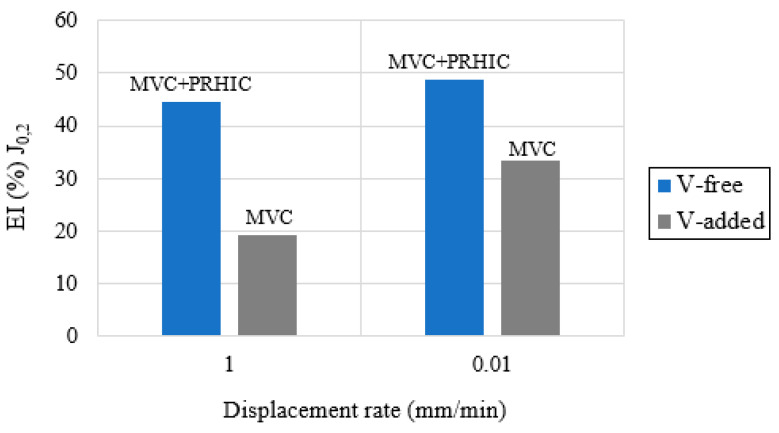
Fracture toughness test results in presence of internal hydrogen at different displacement rates. Within the figure, the identified hydrogen embrittlement mechanisms are also indicated.

**Figure 11 materials-14-07269-f011:**
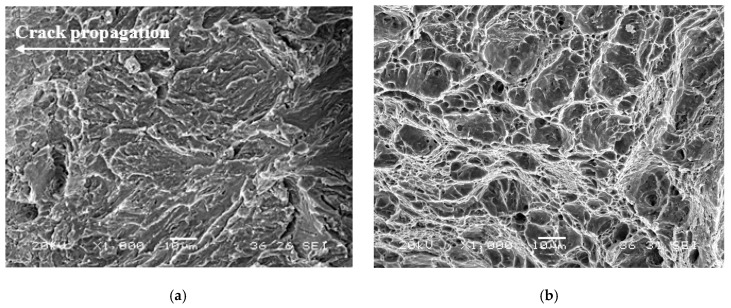
Fracture micromechanisms (near the fatigue precrack front) corresponding to the fracture toughness samples tested with internal hydrogen at the lowest displacement rate, 0.01 mm/min: (**a**) V-free, PRHIC micromechanism; (**b**) V-added, microvoid coalescence (MVC).

**Figure 12 materials-14-07269-f012:**
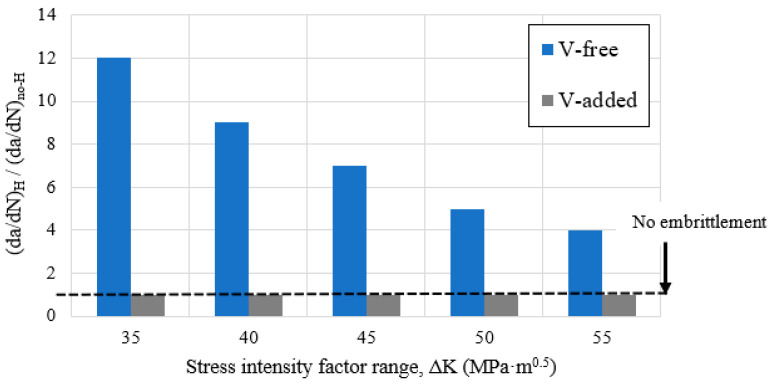
Fatigue crack growth acceleration factor (at 0.1 Hz) determined on the studied steel grades (with R = 0.1) with and without vanadium.

**Figure 13 materials-14-07269-f013:**
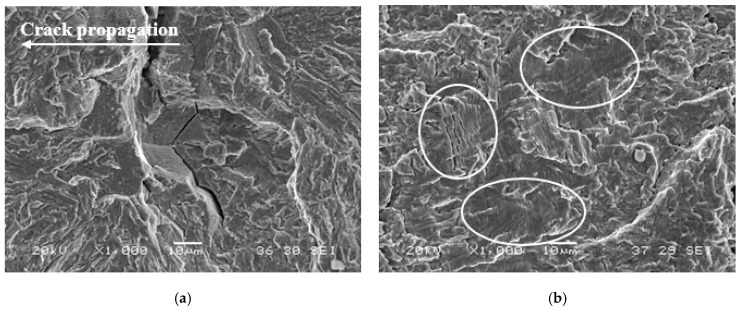
Fracture micromechanisms on the fatigue crack growth rate samples tested in presence of internal hydrogen at 0.1 Hz and R = 0.1: (**a**) V-free, PRHIC with IG fracture around ΔK = 35–40 MPa·m^0.5^; (**b**) V-added, showing ductile striations (white circle).

**Table 1 materials-14-07269-t001:** Chemical composition of V-free and V-added steels (weight %).

Steel Grade	C	Mn	Si	Cr	Mo	Ni	V
V-free	0.14	0.56	0.16	2.23	1.00	0.09	-
V-added	0.15	0.52	0.09	2.27	1.06	0.19	0.31

**Table 2 materials-14-07269-t002:** Applied heat treatments.

Specimen ID	Heat Treatment Sequences
V-free	940 °C/3 h + WQ + 690 °C/30 h tempered + air cooling
V-added	925 °C/90 min + WQ + 720 °C/3 h tempered + air cooling

**Table 3 materials-14-07269-t003:** Hydrogen precharging conditions.

Pressure (MPa)	Temperature (°C)	Time (h)
19.5	450	21

**Table 4 materials-14-07269-t004:** Hardness and tensile properties. ‘σ_Y_’ is the yield strength, ‘σ_t_’ is the ultimate tensile strength, ‘e’ represents the elongation, and HB is the Brinell hardness.

Sample ID	σ_y_ (MPa)	σ_t_ (MPa)	e (%)	HB
V-free	430	580	27	170
V-added	567	714	23	200

**Table 5 materials-14-07269-t005:** Hydrogen content evolution in ppm. H_D_: diffusible hydrogen after long exposure time at RT calculated from $\frac{H_{t=0}-H_{t=72}}{H_{t=0}}$. The hydrogen measurement corresponding to 0 h represents the initial hydrogen content that was introduced into the steels according to the hydrogen precharging methodology.

Sample ID	Exposure Time (in Hours) at Room Temperature				% H_D_
	0	24	48	72	
V-free	0.6	0.2	0.2	0.2	67 (0.4 ppm)
V-added	4.3	4.2	4.1	4.1	5 (0.2 ppm)

**Table 6 materials-14-07269-t006:** Trap activation energies.

Sample ID	Position Peak	−E_a_/R (Slope in Figure 3c)	E_a_ (kJ/mol)	R^2^
V-free	1st	1578	13	0.95
	2nd	2109	18	0.99
V-added	1st	1999	17	0.99
	2nd	4244	35	0.96

**Table 7 materials-14-07269-t007:** Results extracted from the electrochemical hydrogen permeation curves.

Sample ID	* P_app_ (molH/m·s)	C_app_ (ppm)	D_app_ (m^2^/s)
V-free	1.1 × 10^−9^	1.7	1.7 × 10^−10^
V-added	8.8 × 10^−10^	7.1	2.6 × 10^−11^

* Apparent permeability: P_app_ = C_app_ × D_app_.

**Table 8 materials-14-07269-t008:** Results of fracture toughness tests conducted on the uncharged and precharged samples. ‘EI’ represents the embrittlement index, calculated as: EI = [(J − J_H_)/J] × 100.

Sample ID	Displacement Rate (mm/min)	J_0.2_ (kJ/m^2^)	EI (%)	* Fracture Micromechanism
V-free (σ_y_ = 430 MPa)	1 (uncharged)	904	-	MVC
	1 (H-precharged)	502	44.5	MVC + PRHIC
	0.01 (H-precharged)	465	48.6	MVC + PRHIC
V-added (σ_y_ = 567 MPa)	1 (uncharged)	672	-	MVC
	1 (H-precharged)	542	19.3	MVC
	0.01 (H-precharged)	448	33.3	MVC

* MVC: microvoid coalescence; PRHIC: plasticity-related hydrogen-induced cracking [43].

## Data Availability

The data presented in this study are available on request from the corresponding author.

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
