# Peer review of "The Positive Role of Nanometric Molybdenum–Vanadium Carbides in Mitigating Hydrogen Embrittlement in Structural Steels"

_materials, 2021, doi:10.3390/ma14237269_

Round 1
Reviewer 1 Report
L.B.Peral et al. reports an important topic about the Positive Role of The Nanometric Molybdenum-Vanadium Carbides To Mitigate Hydrogen Embrittlement On Structural Steels. The framework is very interesting as the industry is continuously searching for structural steel protection method, especially in harsh condition. The experimental campaign described in the paper is rich and interesting. My overall suggestion is minor revision:
- It is noted that your manuscript needs careful editing by someone with expertise in technical English editing paying particular attention to English grammar, spelling, and sentence structure so that the goals and results of the study are clear to the reader.
- In abstract, some innovative data needs to be added to illustrate the conclusions of this work, such as adhesion, wear rate and corrosion parameters.
- Keyworks: the number of keywork was too many and need to be reduced.
- The introduction section should be improved; some recently published papers about Molybdenum-Vanadium Carbides should be added for comparison and analysis.
- Why chosen structural steel as the substrate? Does it work on other metals?
- Compared with other material, what are the advantages of Nanometric Molybdenum-Vanadium Carbides?
Author Response
First of all, we want to thank the work performed by the reviewers looking to improve the final quality of our manuscript. We have written the modified text in the revised manuscript in red and here, we have included some comments to your questions.
- In abstract, some innovative data needs to be added to illustrate the conclusions of this work, such as adhesion, wear rate and corrosion parameters.
We have modified part of the abstract section in order to illustrate the conclusion part. Parameters such as wear rate or corrosion have not been addressed in this work.
- Keyworks: the number of keywork was too many and need to be reduced.
Yes. The number of keywords has been reduced in the new manuscript.
- The introduction section should be improved; some recently published papers about Molybdenum-Vanadium Carbides should be added for comparison and analysis.
A new paragraph in red color has been added in the introduction section. According to this, new references have been also included in the marked manuscript. In order to compare with other authors, references [40-42] must be also considered, please.
- Why chosen structural steel as the substrate? Does it work on other metals?
Quenched and tempered steels alloyed with chromium, chromium-molybdenum or chromium-molybdenum-vanadium are employed to manufacture vessels and pipelines used to store and transport hydrogen gas. On the other hand, austenitic stainless steels, which are also used in the hydrogen industry, are less sensitive to hydrogen embrittlement but, they are more expensive.
- Compared with other material, what are the advantages of Nanometric Molybdenum-Vanadium Carbides?
Type of trap for hydrogen atoms plays an important role on the hydrogen embrittlement process. In this study, molybdenum-vanadium mixed nanometric carbides (homogeneously distributed within the steel microstructure) act as very strong trapping sites for hydrogen atoms (irreversible trapping character with Ea around 35 kJ/mol). It contributes to retard hydrogen diffusion toward the notch tip region. As a consequence of this, the local hydrogen concentration in the notch tip area is barely able to trigger hydrogen embrittlement micromechanisms (HEDE or HELP). Accordingly, mechanical behaviour of the V-added grade could be notably improved in presence of internal hydrogen.
Reviewer 2 Report
In this manuscript, authors added the Molybdenum and Vanadium carbides into the structural steels, and discussed the positive effect on hydrogen embrittlement. Some properties of structural steels, including tensile properties and hardness, fracture toughness, fatigue crack growth, were measured. The results show that adding V into structural steels can effectively alleviate the hydrogen embrittlement, as well as improve the fracture toughness and fatigue behavior. The logical relationship between the characterization results and the metal properties testing is clear. While the results is interesting, there were still some problems in the manuscript. The words in the images, like Fig. 4-8, have the underline. This should be avoided in the published paper.Thus, I suggest this work can be accept after a minor revision.Author Response
First of all, we want to thank the work performed by the reviewers looking to improve the final quality of our work. We have written the modified text in the revised manuscript in red and here, we have included some comments to your questions.
- The words in the images, like Fig. 4-8, have the underline.
Thank you for your appreciation. We have solved this point in the new marked manuscript.
Reviewer 3 Report
Peral et al. demonstrated the positive role of molybdenum-vanadium nanometric carbides in structural steels. The manuscript is well written and fits into the journal for publication. However, minor revision is required.
- The author needs to bring more discussion in the introduction part to claim the positive side of this materials.
- The author should discuss more on microstructure by bringing more SEM images.
- Fatigue crack growth rate tests. Please add more discussion by referring to other articles.
Author Response
First of all, we want to thank the work performed by the reviewers looking to improve the final quality of our manuscript. We have written the modified text in the revised manuscript in red and here, we have included some comments to your questions.
- The author needs to bring more discussion in the introduction part to claim the positive side of this materials.
New sentences (in red color) have been added in the introduction section. Accordingly, new references have been also included in the marked manuscript to claim the positive effect of these materials (depending on the employed alloying elements for forming carbides) in order to alleviate hydrogen embrittlement.
- The author should discuss more on microstructure by bringing more SEM images.
In this moment, we continue to analyse microstructural singularities related to quenched and tempered CrMoV steel grades, which might have an important effect on hydrogen trapping.
At the moment, we are thinking that the provided information in the microstructural section is enough to explain the mechanical behavior in presence of internal hydrogen. Besides, TDA analysis also corroborates the microstructural singularities displayed in section 3.1.
- Fatigue crack growth rate tests. Please add more discussion by referring to other articles.
More discussion and new references have been added in red color through the marked manuscript.